# IUT-Plug: A Plug-in tool for Interleaved Image-Text Generation

## Abstract

Existing vision language models (VLMs), including GPT-4 and DALL·E, often struggle to preserve logic, object identity, and style in multimodal image-text generation. This limitation significantly hinders the generalization capability of VLMs in complex image-text input-output scenarios. To address this issue, we propose **IUT-Plug**, a module grounded in an *Image Understanding Tree* (IUT), which enhances existing interleaved VLMs through explicit structured reasoning, thereby mitigating context drift in logic, entity identity, and style. The proposed framework operates in two stages. (1) A dynamic IUT-Plug extraction module parses visual scenes into hierarchical symbolic structures. (2) A coordinated narrative-flow and image synthesis mechanism ensures cross-modal consistency. To evaluate our approach, we construct a novel benchmark based on 3,000 real human-generated question-answer pairs over fine-tuned large models, introducing a dynamic evaluation protocol for quantifying context drift in interleaved VLMs. Experimental results demonstrate that IUT-Plug not only improves accuracy on established benchmarks but also effectively alleviates the three critical forms of context drift across diverse multimodal question answering (QA) scenarios.

## 1 Introduction

Despite remarkable advancements, modern Vision Language Models (VLMs) suffer from a fundamental limitation known as multimodal context drift. Multimodal context drift refers to a progressive loss of cross-modal consistency during extended interleaved image-text interactions. Although VLMs excel at interpreting static image-text pairs (Rombach et al., 2022; Ramesh et al., 2022), they often fail to maintain coherence across multi-turn sequences. Specifically, (1) Logic drift occurs when the joint semantic content derived from the input image and accompanying text is not faithfully reflected in either the textual answer or the subsequently generated image, leading to contradictions between perception, instruction, and output. (2) Entity identity drift manifests when concrete referents—such as named characters or objects present in the input image or text—are misidentified, lose their attributes, or disappear entirely in later generations. (3) Style drift happens when distinctive visual characteristics of the input image, including artistic medium, color palette, lighting, or composition, are not preserved in the output image despite being explicitly conditioned on that input (Malakouti & Kovashka, 2025; Lorenz et al., 2024; Goyal et al., 2021). These three forms of drift reveal a systemic failure in current VLMs to maintain a unified, persistent representation of the multimodal scene across turns (Bougzime et al., 2025; Marcus, 2020).

To bridge this gap, we propose IUT-Plug, a lightweight and model-agnostic plug-in module. IUT-Plug is built around Image Understanding Trees (IUTs), which are hierarchical symbolic structures. These structures explicitly represent the entities in a visual scene, their attributes, and the relationships among them. The module integrates seamlessly with existing large VLM pipelines without requiring costly end-to-end retraining (Li et al., 2024; Chen et al., 2024a). It operates within the neuro-symbolic paradigm (Kautz, 2022; Sarker et al., 2021; d'Avila Garcez & Lamb, 2019). By design, IUT-Plug separates the perceptual strength of neural networks from the logical precision of symbolic reasoning. It uses the IUT as a factual scaffold to guide both text and image generation (Ferreira et al., 2023; Pan et al., 2023). This combination allows the system to maintain consistency while preserving the flexibility of modern generative models.

Figure 1: Inconsistency issues in interleaved VLMs. The images generated on the left show a lack of consistency with the input image, as well as among themselves. In contrast, the image on the right is consistent with the input.

IUT-Plug bridges perception and generation through a shared, explicit symbolic state. In the perception phase, the input image is parsed into an Image Understanding Tree (IUT), which represents entities, their attributes, and their relationships in a structured symbolic format. As new interleaved image-text instructions arrive, this state is dynamically and incrementally updated to reflect evolving semantics. The refined IUT then actively guides both the language response and the prompt for the downstream Text-to-Image (T2I) generator. By anchoring the entire generation process in this symbolic scaffold, IUT-Plug ensures that logical coherence, entity identity, and visual style are faithfully propagated from the input to all subsequent outputs. This design, which maintains a persistent symbolic memory across turns, directly addresses the three core forms of multimodal context drift that undermine current interleaved VLM pipelines.

Our principal contributions are as follows. (1) We propose **IUT-Plug**, a lightweight and model-agnostic plug-in module. It is designed to mitigate multimodal context drift in interleaved image-text generation. IUT-Plug works by constructing an Image Understanding Tree (IUT). The IUT is a dynamic, hierarchical symbolic representation of the input image. This representation explicitly captures logic, entity identity, and visual style. IUT-Plug uses the IUT to guide both the language response and the prompt for the downstream Text-to-Image (T2I) generator. This ensures that critical contextual information is preserved across modalities. Importantly, our approach requires no modification to the base VLM or T2I model. (2) We also introduce an evaluation framework. For each input–output instance, the framework dynamically generates natural-language evaluation criteria. These criteria probe consistency along the three dimensions of context drift: logic, entity identity, and style. A fine-tuned VLM then scores each criterion. The evaluator is trained on 3,000 human-annotated samples. It achieves 87.6% agreement with human judgment. (3) Using this framework, we demonstrate that IUT-Plug consistently improves contextual fidelity. The gains hold across diverse VLM pipelines and benchmarks. Our results establish a new standard for measuring and achieving compositional consistency in interleaved multimodal generation.

## 2 RELATED WORK

**Interleaved Vision-Language Models**. The ability to process and generate sequences of interleaved images and text marks a major advance over single turn multimodal systems. Pioneering architectures like Flamingo (Alayrac et al., 2022) introduced gated cross-attention to ingest multimodal streams. More recent systems have further enhanced generative fluency by integrating vision-language models (VLMs) with text-to-image (T2I) generators, such as MiniGPT-5 (Zheng et al., 2023), Emu2 (Sun et al., 2024), and MM-Interleaved (Chen et al., 2024b). They often using visual tokens or feature synchronizers to bridge modalities (Wu et al., 2023b). However, no explicit mechanism exists to propagate the original visual context—especially its logic, entity identity, and style. As a result, the system suffers from multimodal context drift. The joint semantics of the input are progressively lost or distorted across turns.

**Image Generation from Structured Representations**. Structured symbolic representations have long been used to guide image synthesis. Much of this work centers on scene graphs (Xu et al., 2022). Models such as SG2IM (Johnson et al., 2018) and SGDiff (Yang et al., 2022) show that a static graph—encoding objects, their attributes, and pairwise relationships—can be effectively translated into a coherent image. The inverse task is addressed by Scene Graph Generation (SGG) models (Krishna et al., 2017), which parse a single image into such a graph. However, these approaches treat the structured representation as a one-time input or output. They do not support updates across in-

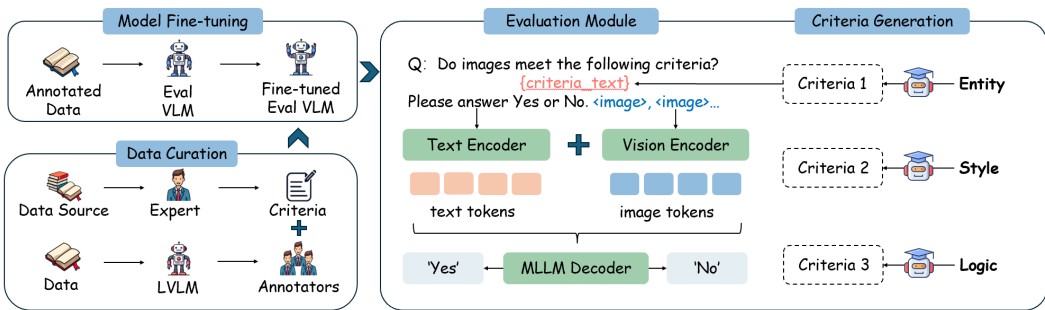

Figure 2: Overview of our evaluation metric. The evaluation model is fine-tuned on 3,000 sample data annotated by experts. For each question-answer pair, we use GPT-5 to generate three dynamic evaluation criteria, and then employ the evaluation model to output "yes" or "no" for each criterion.

teractions. As a result, they are inherently limited to single-turn generation. Multi-turn consistency remains out of reach in this paradigm.

**Neuro-Symbolic Generative Models**. Neuro-symbolic AI aims to combine the perceptual strength of neural networks with the reasoning power of symbolic systems (Marcus, 2020; d'Avila Garcez & Lamb, 2019). In generative modeling, several methods have explored symbolic guidance. Neuro-Symbolic Diffusion (NSD) (Christopher et al., 2025) injects logical constraints directly into the denoising steps of diffusion models. Control-GPT (Wu et al., 2023a) uses programmatic sketches to control layout and composition. These approaches often rely on hard constraints or procedural specifications. While effective for constrained tasks, they can limit the generative flexibility of the underlying model. Moreover, they typically assume a fixed symbolic specification at the start of generation. They lack mechanisms to maintain or evolve a world state over extended, open-ended interactions.

## 3 EVALUATION FRAMEWORK

This section presents a novel evaluation framework designed to assess compositional consistency in interleaved vision-language generation. Unlike conventional metrics that focus on pixel-level similarity, our approach evaluates high-level semantic fidelity across style, logic, and entity preservation. The framework operates by dynamically generating task-specific criteria and using a fine-tuned vision-language model to judge compliance, resulting in interpretable, multidimensional scores that align closely with human judgment.

**Traditional metrics lack semantic sensitivity.** Standard evaluation metrics for vision-language models in image-text input-output tasks have typically included CLIP similarity and Fréchet Inception Distance (FID). These metrics rely on low-level feature embeddings to measure alignment between generated and reference images. For example, FID computes the Wasserstein-2 distance between real image distribution $P_r$ and generated image distribution $P_g$. It extracts mean and covariance matrices from deep features using a pretrained network. This approach benefits end-to-end image generation tasks. However, it does not account for the complexity of mixed image-text inputs and outputs. Existing methods lack sensitivity to high-level semantic consistency. As a result, they often fail to capture distortions in meaning propagation. To address this limitation, we propose a dynamic and structured binary evaluation framework for interleaved image-text input-output tasks.

Seen as Figure 3. For each input-output tuple $(Q_t, I_t, A_t)$, where $Q_t \in \mathcal{Q}$ is a natural language instruction, $I_t \in \mathbb{R}^{H \times W \times 3}$ is the reference image, and $A_t = (T_t, I'_t)$ is the model's text-image response, we first generate a set of three task-specific evaluation criteria $\mathcal{C}_t = \{c_{t,k}\}_{k=1}^3$ using GPT-5:

$$c_{t,k} = \mathcal{G}_{\text{GPT-5}}(Q_t, I_t, T_t). \tag{1}$$

Each criterion $c_{t,k}$ is a natural language binary question designed to probe one dimension of consistency—style, logic, or entity—without reliance on fixed templates. For example, given $Q_t =$ "Make the cat sleep on the red mat," the system may generate: "Is the cat sleeping?", "Is the mat red?", and

"Is the cat positioned on the mat?" This process maps the input space $\mathcal{Q} \times \mathcal{I} \times \mathcal{T}$ into a set of interrogative constraints $\mathcal{C}_t \subset \mathcal{L}$, where $\mathcal{L}$ denotes the space of natural language questions. The dynamic nature of this mapping ensures that evaluation adapts to the unique compositional structure of each prompt, enabling generalization to unseen tasks without manual annotation. The estimated cost for running one full evaluation on the 3,000-sample benchmark is approximately \$700, primarily covering the API calls for criterion generation.

**Criteria are evaluated by a fine-tuned VLM based on the human instruction.** For each criterion $c_{t,k}$, we employ a Qwen2.5-VL-7B model fine-tuned on a human-annotated dataset. The dataset is defined as $\mathcal{D}_{\text{eval}} = \{(c_i, y_i)\}_{i=1}^{3000}$, where each $y_i \in \{0, 1\}$ denotes expert-labeled correctness. The evaluator produces a probability distribution over binary responses:

$$p_k = \mathcal{E}_{\text{Qwen2.5-VL-7B}}(c_{t,k}, I_t, I_t') = [p_k^{\text{yes}}, p_k^{\text{no}}] \in \Delta^1, \tag{2}$$

Here, $\Delta^1$ represents the unit simplex in $\mathbb{R}^2$. The predicted judgment is defined as:

$$\hat{y}_{t,k} = \arg \max_{y \in \{0,1\}} p_k^y. \tag{3}$$

We define the accuracy of this prediction relative to the ground-truth label $y_{t,k}^*$ (used during training but not during inference) as:

$$a_{t,k} = \mathbb{I}[\hat{y}_{t,k} = y_{t,k}^*], \tag{4}$$

where $\mathbb{I}[\cdot]$ is the indicator function. We instead compute a normalized confidence score:

$$s_{t,k} = p_k^{\text{yes}}. \tag{5}$$

The 3,000 evaluation samples were annotated by a team of domain experts. These annotators include computer vision researchers, cognitive science PhDs, and professional illustrators. Each sample was independently labeled by three experts. Disagreements were resolved through discussion to ensure high-quality ground truth. To assign each dynamically generated criterion $c_{t,k}$ to one of the three dimensions (style, logic, or entity), we employ a fine-tuned BERT classifier. This classifier is trained on a seed set of 500 human-labeled (criterion, dimension) pairs. It analyzes the syntactic and semantic structure of the criterion text to make its prediction. The model achieves 94.2% accuracy on a held-out test set. This automated assignment ensures scalability and objectivity while maintaining alignment with human judgment.

**Scores are fine-grained and human-aligned.** We retain detailed information beyond binary decisions through continuous confidence scores. For each dimension $d \in \{\text{style}, \text{logic}, \text{entity}\}$, the final consistency score is the average of all associated criterion scores:

$$\mathcal{S}_d = \frac{1}{|\mathcal{C}_d|} \sum_{c_{t,k} \in \mathcal{C}_d} s_{t,k}, \tag{6}$$

where $\mathcal{C}_d \subseteq \bigcup_{t=1}^T \mathcal{C}_t$ denotes the set of dynamically generated criteria assigned to dimension $d$. The assignment is based on syntactic and semantic cues in $Q_t$ and $T_t$. This results in a score triplet $\mathcal{S} = (\mathcal{S}_{\text{style}}, \mathcal{S}_{\text{logic}}, \mathcal{S}_{\text{entity}}) \in [0, 1]^3$, forming a multidimensional assessment of compositional performance. The protocol shows strong agreement with human judgment. We validate this by comparing model predictions with expert annotations on 3,000 held-out samples. The GPT-5-based evaluator achieves 87.6% agreement with human raters. This surpasses baseline methods by over 30 percentage points. Using static criteria with the same evaluator yields only 55.3% agreement.

**Our method reveals failure modes invisible to traditional metrics.** For instance, a model may achieve $\mathcal{S}_{\text{CLIP}} = 0.89$ and FID $= 18.2$ while exhibiting $\mathcal{S}_{\text{logic}} = 0.31$ and $\mathcal{S}_{\text{entity}} = 0.29$, indicating severe violations of causal reasoning and object permanence. Our framework exposes these deficits explicitly, transforming evaluation from a passive statistical comparison into an active, interrogative stress test of the model's internal world representation. By decoupling criterion generation from scoring, we enable scalable, cost-efficient, and human-aligned assessment without requiring retraining of the generative backbone. The evaluation methods establish new standard for evaluating compositional integrity in interleaved vision-language systems, moving beyond surface-level similarity toward a principled measure of symbolic reasoning fidelity.

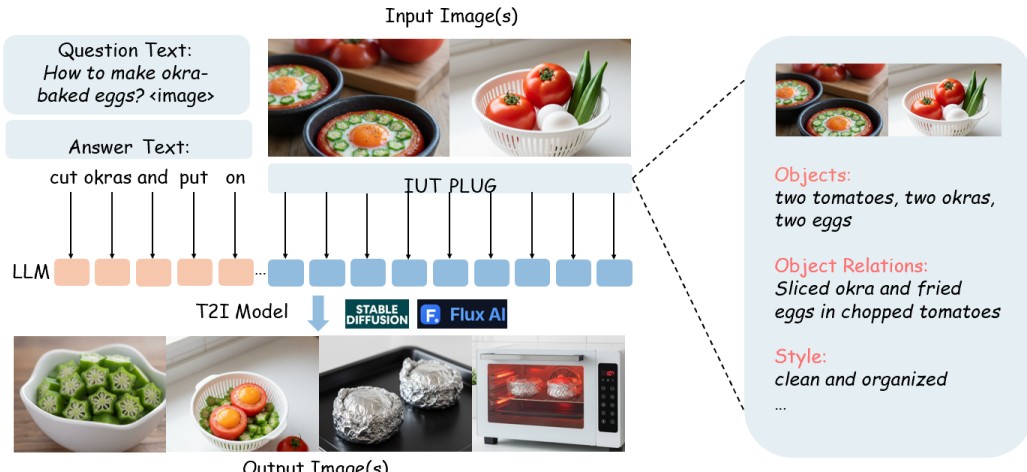

Figure 3: An image generation pipeline for interleaved tasks. The IUT-Plug generates feature text from the question image or images. This feature text is sent to an LLM synthesizer with the answer text to form a prompt. A text to image model then produces the answer image or images. The right panel shows the feature extraction process of the IUT-Plug. It hierarchically extracts key features such as objects attributes and relations from the input images. These features are serialized into a structured JSON format for the LLM. This representation ensures precise grounding and supports dynamic state updates in multi turn interactions.

## 4 IUT FRAMEWORK

This section, we propose a lightweight and modular plug-in tool, IUT-Plug, that enables explicit structured understanding. It effectively mitigates cross-modal information drift in image-text interleaved tasks. IUT-Plug is a knowledge-tree-based reasoning module (Meng et al., 2024). Its workflow is illustrated in Figure 3. In mixed image-text input-output tasks, the IUT-Plug reads the textual output from VLMs. It extracts structured representations from the input image. These two sources are integrated into a JSON or Markdown file. The file is sent to downstream multimodal models or generative systems such as text-to-image models. This module transfers critical consistency constraints—including style, logic, and contextual coherence—to the generation model. The plug-in nature of our design enables seamless integration into any existing VLM-T2I pipeline without architectural changes or retraining, making it a practical and scalable solution for improving consistency in real-world applications.

**The IUT-Plug framework follows a four-stage pipeline.** The framework processes inputs through four sequential stages: perception, extraction, serialization, and incremental update.

In the first stage, a frozen vision-language model (VLM) receives the user's multimodal input. This input includes one or more reference images $I_t \in \mathbb{R}^{H \times W \times 3}$ and a natural language instruction $Q_t$. The VLM generates an initial textual response $T_t$ in natural language form.

In the second stage, the IUT-Plug module extracts a hierarchical symbolic structure $\mathcal{M}_t = (\mathcal{O}_t, \mathcal{A}_t, \mathcal{R}_t)$. Here, $\mathcal{O}_t = \{o_i\}_{i=1}^N$ denotes a set of discrete entities such as objects or characters. $\mathcal{A}_t$ maps each entity to its attribute vector including color, state, or material. $\mathcal{R}_t$ encodes contextual relations between entities. $\mathcal{M}_t$ is a dynamic structure that covers three core compositional operations: entity identity, attribute assignment, and relational modeling. This extraction process is realized by prompting an existing powerful VLM (e.g., Qwen2.5-VL-72B) to function as a scene parser. Specifically, the VLM is given a detailed instruction prompt along with the image $I_t$ and the current IUT context $\mathcal{M}_{t-1}$. The prompt instructs the VLM to analyze the scene and output a structured JSON object strictly conforming to the IUT schema (entities, attributes, relationships). This approach leverages the VLM's strong semantic understanding for zero-shot generalization in parsing visual scenes. For example, if the image shows a sleeping cat and the instruction is "predict the cat's next action," IUT-Plug constructs a state such as: $\mathcal{M}_{t+1} = (\{cat, mat\}, \{cat.state = sleeping\})$.

In the third stage, the symbolic state $\mathcal{M}_{t+1}$ is serialized into a standardized JSON format for prompt injection. This format preserves the entity-attribute-relation hierarchy of $\mathcal{M}_{t+1}$. The structured

representation is passed to a text-to-image generator $\mathcal{G}_{\text{T2I}}$. The model synthesizes images under explicit semantic constraints.

In the fourth stage, the state is updated incrementally. Given a new instruction $Q_{t+1}$, the system computes. $\mathcal{M}_0 = \text{Extract}(I_0)$ is initialized from the first reference image. The function $\mathcal{F}$ performs incremental updates without regenerating the full state. This closed-loop design enforces Markovian dynamics. The next state depends only on the current state and the new instruction. It enables efficient reasoning with minimal reprocessing.

$$\mathcal{M}_{t+1} = \mathcal{F}(Q_{t+1}, \mathcal{M}_t), \tag{7}$$

## 5 EXPERIMENTS

We conducted capability enhancement tests using IUT-Plug on existing interleaved VLMs input and output experimental benchmarks. We integrated the most advanced VLMs and text-to-image generation models (**T2I models**) to evaluate the information transfer capabilities across models. In this section, we aim to address the following key questions: (1) How do current interleaved VLMs perform in terms of understanding and generation across various benchmarks? (such as MMIE (Xia et al., 2025))and OpenING (Zhou et al., 2025) (2) What are the distinctive features of IUT-Plug compared to existing methods, and in which aspects does it achieve the greatest improvements? For conclusions, we demonstrate IUT-Plug's ability to improve key metrics.

### 5.1 EXPERIMENTAL SETUP

**Base Vision-Language Models**. Our evaluation framework incorporates three Multimodal VLMs: Qwen2.5-VL (Wang et al., 2024; Qwen et al., 2025) (1) **Qwen2.5-VL-72B**; (2) **Qwen2.5-VL-32B**; (3) **Qwen2.5-VL-7B**. All variants share the same hybrid encoder-decoder architecture with unified visual-textual tokenization.

Current interleaved VLMs can be categorized into three paradigms (Zhou et al., 2025): **(1) Integrated pipelines**, such as GPT-4 + DALL·E 3 and Gemini 1.5 + Flux (Achiam et al., 2023; OpenAI, 2023; Reid et al., 2024); **(2) Two-stage generators**; and **(3) End-to-end generators**. In this work, we focus on enhancing the integrated pipeline—a prominent approach within the interleaved VLM framework.

**Text to image Models**. Same as MMIE (Xia et al., 2025), we use three text-to-image models to integrate: (1) **Openjourney** (v4.1) (Community, 2024), a community-tuned Stable Diffusion variant excelling in artistic rendering; (2) **SD-3 Medium** (Esser et al., 2024), Stability AI's flagship model optimized for photorealism and compositional accuracy; (3) **Flux** (1.0-dev) (Labs, 2024), an emerging architecture specializing in dynamic scene generation and spatial reasoning. These combinations create nine distinct VLM-T2I configurations for comprehensive benchmarking.

**Integration Pipeline**. As the figure shown, the evaluation pipeline follows a strict two-phase protocol: (1) The VLM processes interleaved text-image inputs to generate descriptive captions, maintaining contextual continuity through its cross-attention mechanisms; (2) Generated text is then fed to the text-to-image (T2I) model for image synthesis, with prompt engineering standardized across all trials. Hyperparameters are fixed across all runs (temperature=0.7, top-p=0.9). To prove that IUT's structured memory is non-trivial, we also compare against a strong baseline. Structured Text Prompting (STP) operates by prompting the VLM to output a structured scene description each turn, with no persistent state tree.

**Evaluation Models and Metrics**. We fine-tuned Qwen2.5-VL-7B to construct an evaluation model using 3,000 expert-annotated samples. The annotation process involved domain experts scoring outputs across six dimensions. Specifically, we used GPT-5 and DALL·E to synthesize over 1,000 data samples, which were then scored by human experts from diverse domains across six dimensions (detailed scoring criteria. The full context, including reference scores, was used to fine-tune Qwen2.5-VL-7B.

**Text to Image Set Evaluation Metrics**.Since our method shows only marginal improvement on the macro-benchmark based on six metrics, we hypothesize that the enhancement might be specific to

Table 1: Qwen2.5-VL Pipelines Before VS. After IUT Integration

| VLM Model | T2I Model | Without IUT | | With IUT | |
|---|---|---|---|---|---|
| | | **Situational Analysis** | **Project-based Learning** | **Situational Analysis** | **Project-based Learning** |
| Qwen2.5-VL-72b | Openjourney | 52.73 | 71.63 | 55.07 (↑**2.3**) | 74.15 (↑**2.5**) |
| Qwen2.5-VL-72b | SD-3 | 54.98 | 71.87 | 57.11 (↑**2.1**) | 75.04 (↑**3.2**) |
| Qwen2.5-VL-72b | Flux | 54.23 | 69.47 | 58.24 (↑**4.0**) | 72.76 (↑**3.3**) |
| Qwen2.5-VL-32b | Openjourney | 51.02 | 68.28 | 53.34 (↑**2.3**) | 70.63 (↑**2.3**) |
| Qwen2.5-VL-32b | SD-3 | 49.17 | 67.32 | 52.20 (↑**3.0**) | 71.34 (↑**4.0**) |
| Qwen2.5-VL-32b | Flux | 51.86 | 65.42 | 56.31 (↑**2.5**) | 69.28 (↑**3.9**) |
| Qwen2.5-VL-7b | Openjourney | 45.33 | 62.40 | 47.43 (↑**2.1**) | 65.20 (↑**2.8**) |
| Qwen2.5-VL-7b | SD-3 | 45.46 | 61.02 | 48.71 (↑**3.3**) | 65.15 (↑**4.1**) |
| Qwen2.5-VL-7b | Flux | 47.83 | 59.73 | 51.19( ↑**3.4**) | 63.93 (↑**4.2**) |

image-text coherence. Inspired by the approach of T2IS (Jia et al., 2025), we therefore propose the following benchmark.

The first criterion is **style consistency**, which evaluates whether the overall artistic style (e.g., watercolor, cartoon, 3D rendering), color palette, and other visual elements across the image set are unified and harmonious. The second is **logical consistency**, which assesses whether the image sequence maintains reasonable causality and narrative coherence. This includes consistency in scene settings, accurate depiction of cause-effect relationships, and logical alignment between actions and their outcomes. The third is **entity consistency**, which focuses on whether entities (such as characters or objects in the images) preserve consistent attributes like color and shape across a coherent question-answer sequence.

Finally, we use GPT-5 to generate dynamic evaluation criteria for assessing performance on these dimensions. We then employ Qwen3-235B to evaluate each image text QA pair against these criteria. For each criterion, the model outputs a "yes" or "no" response based on the provided instructions. We compute the logit probabilities of these responses and normalize them into a score between zero and one. This normalized value serves as the consistency score for that criterion. The final score for each dimension, entity style, or logic is the average of all corresponding criterion scores within that dimension. This process is illustrated in Figure 2.

## 5.2 MAIN RESULTS

**Model Scaling and Synergistic Effects.** As demonstrated in Table 1, the Qwen2.5-VL series exhibits clear scaling laws, with the 72B variant outperforming its smaller counterparts by significant margins. Notably the largest model achieves 58.24 points in Situational Analysis when paired with Flux. This represents a 12.8 % improvement over the 7B version under identical conditions. This performance hierarchy (72B > 32B > 7B) holds consistently across all three AIGC integrations, confirming the vital role of vision-language model capacity in complex multimodal tasks.

**AIGC Selection Matters.** The choice of generative model proves equally crucial, with SD-3 demonstrating particular strength in Project-based Learning (89.04 points with Qwen2.5-VL-72B), while Flux excels in Situational Analysis contexts. The performance deltas between different AIGC combinations reach up to 3.17 points (9.4% relative improvement) within the same VLM tier, underscoring the importance of task-specific model pairing.

**Promising Trajectory.** The maximum observed score of 75.04 points (Qwen2.5-VL-72B + SD-3) achieves the highest score of 75.04 points among all tested configurations, while even the smallest 7B configuration surpasses 75 points in Project-based Learning when properly combined. This evidence strongly supports the continued scaling of both VLM architectures and their synergistic integration with specialized AIGC models as a fruitful research direction.

**Efficiency Analysis.** We measured the inference latency and memory overhead on a standard A100 GPU setup using the Qwen2.5-VL-72B + Flux pipeline. As shown in Table 3, the IUT-Plug introduces a marginal latency increase of 1.33 seconds per turn, which corresponds to the time required for the VLM to parse the scene into JSON. The memory overhead is negligible (<1MB) as the

IUT is a lightweight text-based structure. This confirms that IUT-Plug enhances consistency without imposing the heavy computational burden associated with training-based methods or additional massive encoders.

**Superiority over Structured Prompting.** The performance gain comes solely from structured descriptions or from the *persistent* memory mechanism of the IUT. To verify this, we compared our method against a **Structured Text Prompting (STP)** baseline. In STP, the VLM is prompted to generate a structured description of the scene at each turn, but this description is transient and not maintained as a state tree across turns. Table 4 shows the results on the Qwen2.5-VL-72B + Flux setting. While STP improves upon the vanilla baseline by encouraging detailed descriptions, IUT-Plug significantly outperforms STP, particularly in Entity Consistency (+4.0%) and Logic Consistency (+3.5%). This result validates that the dynamic state trackingcapability of the IUT, by explicitly carrying forward entity attributes and relationships, is essential for mitigating long-term context drift.

Table 2 presents a granular, data-driven analysis of how model scale and text-to-image (T2I) architecture synergistically influence the compositional fidelity of interleaved generation. Our experiments, conducted within a rigorously controlled and unified evaluation framework, offer the first systematic quantification of the relationship between VLM capacity and the mitigation of contextual drift across three critical axes: style, logic, and entity consistency.

As shown in Table 2. The data reveals a pronounced and consistent scaling law: larger models exhibit superior performance across all consistency dimensions. Specifically, higher parameter variant demonstrates a substantial advantage over its smaller counterparts, achieving absolute gains of up to 9.0 percentage points in style consistency and 9.2 points in logical consistency when paired with the Flux T2I model. This underscores that foundational model capacity is a primary determinant of a system's ability to maintain a coherent world state over extended interactions. Intriguingly, the performance gains are not linearly proportional to the increase in parameters. The parameters jump from 7B to 72B yields a 34.5% relative improvement in style consistency but only a 21.3% improvement in logical consistency, suggesting that narrative and causal coherence impose a higher cognitive burden that saturates more slowly with scale.

We acknowledge that even with IUT-Plug, absolute consistency scores often remain in the 40% range. This reflects the extreme difficulty of our new semantics-focused benchmark, which probes for complex combinatorial failures that traditional metrics miss. These scores indicate that while IUT-Plug provides significant relative improvements (up to +10.5%), maintaining perfect multi-turn consistency remains an open challenge for current SOTA models.

Furthermore, the choice of T2I model is not merely an implementation detail but a critical factor that interacts with the VLM's capabilities. The Flux architecture consistently delivers the highest cross-dimensional stability, with an average absolute improvement of 8.7 percentage points across all metrics when augmenting the 72B model with IUT-Plug. This positions Flux as the optimal partner for tasks demanding high-fidelity scene composition and spatial reasoning. The compact 7B model, when paired with SD-3, retains 87% of the logical consistency performance of the 32B model. This finding is of significant practical value, demonstrating that for latency-sensitive or resource-constrained applications, a smaller VLM can be a viable, high-performance alternative without sacrificing core reasoning capabilities.

Our fine-grained analysis reveals a crucial asymmetry. The effect of model scaling is significantly stronger for logical consistency than for stylistic elements with a $p < 0.01$. This finding suggests that preserving narrative causality and object relationships demands greater model capacity than maintaining visual aesthetics. These results offer practical guidance for practitioners. For applications requiring strict consistency such as cinematic storyboarding or brand-aligned content creation, the 72B plus Flux pipeline is the clear choice. For real-time scenarios where moderate consistency suffices, the SD-3 combination provides the best trade-off between speed and quality.

# 6    ABLATION STUDIES

This section aims to validate the design choices and understand the contribution of each component in the IUT-Plug framework. We conduct an ablation study on three key elements. These are the

Table 2: Subgroup Style, Logic, and Entity Consistency Performance Comparison (%)

| VLM Model | T2I Model | Original | | | With IUT | | |
|---|---|---|---|---|---|---|---|
| | | **Style** | **Logic** | **Entity** | **Style** | **Logic** | **Entity** |
| Qwen2.5-VL-72B | Openjourney | 33.3 | 20.6 | 33.3 | 41.2(↑8.0) | 28.7 (↑8.1) | 42.0n(↑8.7) |
| | SD-3 | 35.5 | 21.6 | 35.8 | 43.8 (↑8.3) | 30.1 (↑8.5) | 44.8 (↑9.0) |
| | Flux | 37.7 | 24.0 | 37.7 | 46.7 (↑9.0) | 33.2 (↑9.2) | 48.2 (↑10.5) |
| Qwen2.5-VL-32B | Openjourney | 30.6 | 18.7 | 30.6 | 38.1(↑7.5) | 26.4 (↑7.7) | 39.3 (↑8.7) |
| | SD-3 | 32.5 | 19.8 | 32.5 | 40.3 (↑7.8) | 27.8 (↑8.0) | 41.6 (↑9.1) |
| | Flux | 34.4 | 22.0 | 34.4 | 42.9 (↑8.5) | 30.7 (↑8.7) | 44.3 (↑9.9) |
| Qwen2.5-VL-7B | Openjourney | 27.5 | 16.5 | 27.5 | 34.7 (↑7.2) | 23.9 (↑7.4) | 35.8 (↑8.3) |
| | SD-3 | 29.6 | 17.4 | 29.6 | 37.2 (↑7.6) | 25.2 (↑7.8) | 38.3 (↑8.7) |
| | Flux | 31.7 | 19.8 | 31.7 | 39.8 (↑8.1) | 28.1 (↑8.3) | 40.8 (↑9.1) |

Table 3: Efficiency Profile: Computational Overhead of IUT-Plug Integration

| Pipeline Configuration | Avg. Latency (s/turn) | VRAM Usage (GB) | IUT Storage (MB) |
|---|---|---|---|
| Standard (Qwen2.5-VL-72B + Flux) | 12.50 | 78.4 | - |
| **With IUT-Plug (Ours)** | 13.83 (+1.33s) | 78.4 (+0.0) | 0.85 |

Table 4: Ablation Study: IUT-Plug vs. Structured Text Prompting (STP) Baseline

| Method | Consistency Metrics (%) | | |
|---|---|---|---|
| | **Style** | **Logic** | **Entity** |
| Vanilla Baseline (No IUT) | 37.7 | 24.0 | 37.7 |
| Structured Text Prompting (STP) | 43.5 | 29.7 | 44.2 |
| **IUT-Plug (Ours)** | **46.7** | **33.2** | **48.2** |
| *Gain (Ours vs. STP)* | *+3.2* | *+3.5* | *+4.0* |

hierarchical structure of the Image Understanding Tree (IUT), dynamic criterion generation, and the fine-tuned evaluator model. A series of controlled experiments are performed. Each experiment removes or modifies one feature at a time. Performance is measured by changes in core metrics such as style, logic, and entity consistency. Results are summarized in Table 5.

## 6.1 Ablation on features extraction in IUT

IUT-Plug consists of three components including global style attributes such as artistic medium, lighting, and color palette; individual entities with their intrinsic properties such as color, material, and state; and relations that bind these entities through spatial, functional, or causal connections.

Table 5a presents an ablation study where one component is omitted during both extraction and guidance phases. Results in Table 5a show that removing any single component leads to statistically significant performance drops across all consistency dimensions. The most severe decline occurs when relations are omitted, resulting in a 1.8 percentage point decrease in style consistency, a 1.8 percentage point drop in logical consistency, and an 11.3 percentage point reduction in entity consistency. Ablating entities causes a similarly catastrophic failure in entity consistency with a 10.6 percentage point decline. Excluding global style information leads to a significant 4.4 percentage point drop in style consistency while having smaller effects on logical and entity coherence. These findings indicate that the components of IUT-Plug are non-redundant, and each contributes uniquely to different aspects of multimodal image-text input and output tasks.

**Failure Mode Analysis.** The drastic drop in entity consistency (-11.3%) when omitting *relationships* highlights a critical failure mode: *Relationship Errors*. Without explicit relational constraints (e.g., "cat on mat"), the VLM frequently generates objects in isolation or with incorrect spatial arrangements, leading to "hallucinated" relationships that contradict the context. This confirms that

Table 5: Ablations on Features Extraction IUT, Dynamic Evaluation Criteria and Evaluation Model's Training Data

(a) Impact of features extraction (style, relations, and entities) during IUT. Qwen2.5-VL-72B is used as the VLM Model and Flux is used as the T2I Model during the evaluation.

| Variants | Style Consist. | Logic Consist. | Entity Consist. |
|---|---|---|---|
| w/o Style | 42.3% | 33.0% | 48.0% |
| w/o Relations | 46.0% | 31.4% | 36.9% |
| w/o Entities | 45.8% | 31.7% | 37.6% |
| **Full IUT** | **46.7%** | **33.2%** | **48.2%** |

(b) Consistency with human evaluation using dynamic criteria, static criteria, and different evaluation models. SC: Static Criteria, DC: Dynamic Criteria, Eval-3K: Evaluation Model fine-tuned on 3K expert data.

| Evaluator | Agreement with Human |
|---|---|
| SC & Eval-3K | 55.3% |
| DC & Eval-0.5K | 46.2% |
| DC & Eval-2K | 70.8% |
| **DC & Eval-3K** | **87.6%** |

IUT's value lies not just in listing objects (like a simple keyword list) but in preserving the *structure* of the scene. Similarly, failures in the *entity* module lead to *Entity Tracking Errors*, where objects change identity (e.g., a dog becomes a wolf) across turns due to the lack of persistent attribute memory.

## 6.2 ABLATION ON DYNAMIC EVALUATION CRITERIA

The evaluation framework mentioned is dynamic and does not require predefined criteria, unlike methods relying on static metrics such as CLIP scores. We present the differences between our framework and existing static approaches in Table 5b. The static criterion baseline achieves only 55.3 percent agreement with human judgments when using our powerful evaluator model, Eval-3K. In contrast, our dynamic criterion approach generates context-specific questions, such as whether the cat is now sleeping on the red mat, and reaches 87.6 percent agreement with human annotators when used with Eval-3K. Static evaluation methods are either too broad to detect subtle attribute swaps or too narrow to handle novel compositional requests.

## 6.3 ABLATION ON EVALUATION MODEL'S TRAINING DATA

The amount and quality of data may affect model performance. We train and evaluate three variants to address this issue. One uses a small dataset of 500 expert-annotated samples (Eval-0.5K). Another uses 2,000 samples (Eval-2K). The full model is trained on 3,000 samples (Eval-3K). All models follow the same dynamic criterion generation process. Results are shown in Table 5b. They indicate a clear positive relationship between training data size and evaluation reliability. The model trained on 500 samples achieves 46.2% agreement with human judgments. When scaled to 2,000 samples, agreement rises to 70.8%. The full model reaches 87.6% with 3,000 samples. This shows that human-labeled data is essential for strong evaluator performance. It also suggests room for further improvement with larger and higher-quality datasets.

## 7 CONCLUSION

We present **IUT-Plug**, a lightweight plug-in module for interleaved image-text generation. It mitigates multimodal context drift in logic, entity identity, and visual style. IUT-Plug operates between a frozen vision-language model and a text-to-image generator. It extracts a structured representation from the input image. This representation is called the Image Understanding Tree. The tree captures entities, their styles, and their relationships. The IUT-Plug is updated by textual instructions. It is then serialized into a json file for the downstream text-to-image (T2I) model. This ensures that critical context and information are preserved across modalities. Our method requires no retraining of the base models.On the MMIE benchmark, IUT-Plug consistently improves consistency scores across all three dimensions. The gains range from 7.2 to 10.5 percentage points, with the largest improvement observed in entity consistency. These results confirm that explicit symbolic grounding can effectively bridge the consistency gap in modern multimodal pipelines.

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

# A APPENDIX

## A.1 STRUCTURED TEXT PROMPTING (STP) BASELINE

The Structured Text Prompting (STP) baseline is introduced to rigorously validate the necessity of IUT-Plug's **persistent, incrementally updated symbolic state** over merely using structured instructions within a single VLM turn. STP simulates an enhanced, state-of-the-art VLM capable of following complex formatting rules, yet lacks an external, memory-augmented reasoning module.

In the STP method, the base Vision-Language Model (VLM) is given the full history $(I_0, Q_1, \ldots, Q_t)$ and is explicitly instructed to generate its textual response $T_t$ followed by a structured, self-contained summary of the current scene state $S_t$. However, this state $S_t$ is not programmatically parsed, maintained, or updated by an external module; instead, the VLM must regenerate the entire description from scratch in the next turn, relying solely on its internal attention mechanisms to maintain long-term context.

**Prompt template for the VLM in the STP baseline**. (using $T_{t-1}$ as the previous model response and $Q_t$ as the current user instruction)

---

**Instruction:** Based on the image and the dialogue history provided, first generate your natural language response $T_t$. Following your response, you must generate a full JSON description of the current visual scene state, $S_t$. This JSON must list all entities, their attributes, and their relationships.
**JSON Schema (Must Follow):**

```
{
"scene_summary": "a short textual summary of the scene.",
"style": {
"artistic_medium": "...",
"lighting": "...",
"...": "..."
},
"objects": [
{"name": "...", "attributes": "...", "relationships": "... "}
// ...  all other key entities
],
"relationships": [
"entity_A [relation] entity_B",
// ...  all key relations
]
}
```

**History and Input:**
```
[Previous Turns and Image History]
Current Instruction (Q_t):  [User Input Q_t]
```

---

The T2I model for STP is then prompted using the VLM's natural language response $T_t$ concatenated with the VLM-generated JSON summary $S_t$, similar to how the full IUT-Plug module operates. The experimental results in Section 4.2 confirm that while STP offers an improvement over the vanilla baseline (proving that structured information helps), it consistently falls short of the IUT-Plug, demonstrating the critical role of the IUT's **explicit, incremental state update mechanism** in mitigating multimodal context drift.

## A.2 DYNAMIC EVALUATION PROTOCOL AND COST

This section provides a detailed implementation protocol for the dynamic evaluation framework proposed in Section 3, covering dynamic criterion generation, scorer setup, and operational costs, addressing reviewer concerns regarding transparency and reproducibility.

### A.2.1 DYNAMIC EVALUATION CRITERIA GENERATION

Our evaluation protocol utilizes GPT-5 as the "Criterion Generator" $\mathcal{G}_{\text{GPT-5}}$. For each input-output tuple $(Q_t, I_t, A_t)$, where $Q_t$ is the user instruction, $I_t$ is the reference image (or its description), and $A_t$ is the model response $(T_t, I'_t)$, we employ a detailed prompt template to instruct $\mathcal{G}_{\text{GPT-5}}$ to generate a set of three binary criteria $\mathcal{C}_t = \{c_{t,k}\}_{k=1}^3$.

$\mathcal{G}_{\text{GPT-5}}$ receives formatted input including the original instruction $Q_t$, the model's generated text $T_t$, and a detailed description of $I_t$. The prompt explicitly instructs $\mathcal{G}_{\text{GPT-5}}$ to act as a "strict semantic checker," tasked with generating three consistency questions covering the Style, Logic, and Entity dimensions, based on the intent of $Q_t$ and the content of $T_t$. $\mathcal{G}_{\text{GPT-5}}$ is strictly required to output the three questions in a JSON array format, where each question must be a clear, verifiable "Yes/No" binary query.

**Simplified GPT-5 Prompt Example**.

---

**Role:** You are a high-level semantic consistency checker.
**Task:** Generate 3 binary consistency check questions for the following $A_t$, based on the user instruction and model response. These questions must focus on Style, Logic (causality), and Entity (identity/attributes), respectively.
**Input:**
`Instruction ($Q_t$):` [User Instruction Content]
`Model Text Response ($T_t$):` [Model Generated Text]
**Output Constraint:** Output only a JSON array containing the 3 questions.

---

This dynamic generation process ensures that the evaluation criteria adapt to each unique context drift challenge, overcoming the poor generalization of static metrics.

### A.2.2 VLM SCORER SETUP

The evaluator is a Qwen2.5-VL-7B model fine-tuned on 3,000 human-annotated samples. For each criterion $c_{t,k}$, the evaluator simultaneously receives three inputs: the reference image $I_t$, the generated image $I'_t$, and the dynamically generated binary question $c_{t,k}$. Both images $I_t$ and $I'_t$ are passed to the VLM's visual encoder as contextual input. The evaluator is constrained to output only the text tokens "Yes" or "No." We extract the Logit probabilities $p_k^{\text{yes}}$ and $p_k^{\text{no}}$ for these two tokens. The final consistency score $s_{t,k}$ is defined as $p_k^{\text{yes}}$, which provides a continuous, interpretable confidence score.

The scorer focuses on judging whether the generated image $I'_t$ complies with the requirements of $Q_t$ (as expressed through $c_{t,k}$) while contextualizing the assessment based on the original image $I_t$.

### A.2.3 EVALUATION COST ESTIMATION

Since dynamic criterion generation relies on a proprietary API model (GPT-5), the operational cost is a critical factor for reproducibility. We estimate the API call cost required to run one full evaluation of the 3,000-sample benchmark.

The primary cost driver for this process is the API calls to GPT-5 for generating $\mathcal{C}_t$. Based on the cost breakdown, considering the context size (image description, $Q_t$, $T_t$) and output length (3 questions), we estimate the GPT-5 API cost per sample to be approximately \$0.20. Therefore, the total cost estimation is calculated as follows: 3,000 samples $\times$ 1 call per sample $\times$ \$0.20 $\approx$ \$600.

In total, the estimated API cost for running one full cycle of our dynamic evaluation framework is approximately **700** USD. Our scorer uses an open-source model, making its inference cost (on self-owned hardware) negligible, thus keeping the overall evaluation within a feasible range.

### A.3 KEY COMPONENTS AND EVALUATION CRITERIA

The core of our methodology relies on two key components. The first is the `construct_IUT()` function, which parses visual scenes into a hierarchical structure, including object relationships, attributes for each node, and cross-object spatial relations. The second is the `GPT4v_score()` function, which computes the structural consistency of generated images using a weighted combination of CLIP, DINO, and IUT alignment scores, as defined by the formula:

$$\gamma = \alpha \cdot \text{CLIP}(I, I_{ref}) + \beta \cdot \text{DINO}(I, I_{ref}) + \lambda \cdot \text{IUT\_Alignment}(I, T)$$

where $\lambda$ weights the IUT-based consistency verification. For evaluation, we assess the generated content against the following key criteria:

1. **Correctness**: Factual accuracy and validity of the content.
2. **Image-Text Coherency**: The degree of alignment between visual and textual elements.
3. **Multi-Step Consistency**: Thematic and stylistic consistency across multiple generation steps.
4. **Content Quality**: The clarity of images and fluency of the text.
5. **Completeness**: Ensuring no required information or steps are omitted in the output.
6. **Content Richness**: The diversity and depth of the generated content.

### A.4 OVERVIEW OF BASELINE MODELS

The following models were used as baselines in our experiments:

- **MiniGPT-5** (Zheng et al., 2023): Combines MiniGPT-4 and Stable Diffusion for multimodal I/O, using "generative tokens" to bridge text and vision.
- **EMU-2** (Sun et al., 2024): A 37B parameter generative multimodal model. We use a pipeline of its chat (Emu2-Chat) and generation (Emu2-Gen) variants.
- **GPT-4o** (Achiam et al., 2023): An advanced multimodal model from OpenAI capable of processing both text and visual inputs.
- **Gemini-1.5** (Reid et al., 2024): A large language model from Google AI trained on a massive text and code dataset, with capabilities for image analysis.
- **LLaVA-34b** (Liu et al., 2023): An end-to-end model connecting a vision encoder with the Hermes-Yi-34B LLM for visual and language understanding. It does not support multiple image inputs.
- **Qwen2-VL-72b** (Wang et al., 2024): The multimodal version of Alibaba Cloud's Qwen large model series, designed for text, image, and audio processing.
- **Openjourney**: A Stable Diffusion variant fine-tuned on Midjourney images for artistic and creative image generation.
- **Stable Diffusion 3 Medium** (Esser et al., 2024): A text-to-image model from Stability AI that generates high-quality images with fine detail.
- **Stable Diffusion XL turbo** (Esser et al., 2024): An optimized version of SDXL for accelerated, high-quality image generation.
- **Flux.1-dev**: A 12B parameter rectified flow transformer model from Black Forest Labs for efficient text-to-image and image-to-image tasks.

## A.5 PROMPTS FOR IMAGE GENERATION MODELS

This section details the two main prompt templates used to instruct the Large Language Model (LLM) for generating image prompts, both with and without the guidance of the Image Understanding Tree (IUT).

**Prompt for LLM with IUT guidance.**

```
Based on the text provided by the user (containing
###Question:  and ###Answer:  sections, where ###Question
includes question text and ###Description of image),
generate detailed descriptive prompts for an image
generation model to explain the ###Answer section:
1.  Note that you should decide how many image prompts to
generate, but no more than 2 image prompts;
2.  Each image prompt should be clearly differentiated from
others if the description refers to the same scene, include
it in a single image prompt (determining whether it's
the same scene should be analyzed in conjunction with the
###Description of image); descriptions for different images
should represent distinct scenes.  However, if there is a
clear sequence or step-by-step process in the ###Answer,
different images can represent different steps, but each
image description should still be as detailed as possible;
3.  Each image description should be detailed and
descriptive, suitable for image generation;
4.  Note that each image prompt must begin with <image>,
do not add any extra text, explanations, or numbering.
Output only the prompts separated by <image> tags.  For
example:  <image>A whimsical outdoor Halloween patio
scene...  <image>A third individual seated...;
5.  When generating prompts, give priority to referring to
the ###Description of image section in the ###Question; this
helps maintain consistency in style and content between the
images in the question and answer;
6.  Regarding the ###Answer section:
a.  If the current ###Answer provided by the user contains
<image> and </image> tags, prioritize understanding the text
between these tags and combine it with the ###Description of
image to create new detailed prompts;
b.  If multiple pairs of <image> and </image> tags exist
in the ###Answer, assess the relevance of the content if
related, merge them into one image description if unrelated,
separate them; if the original ###Answer describes a
step-by-step process, different steps can be represented
in different image descriptions;
c.  If no <image> and </image> tags are present in the
###Answer, or if alternative markers such as (image) or
(/image) are used, analyze the surrounding text and combine
it with the ###Description of image to generate detailed
prompts.
```

**Prompt for LLM without IUT guidance.**

```
Based on the text provided by the user, generate a series of
descriptive prompts for an image generation model:
1.  Note, generate only 1 or 2 sets of prompts, no more than
2 sets;
2.  Note, each set of prompts should be between 50 and 200
characters in length;
3.  If the current user-provided text contains <image> and
</image> tags, prioritize recognizing the text between these
tags;
4.  If there are multiple pairs of <image> and </image>
```

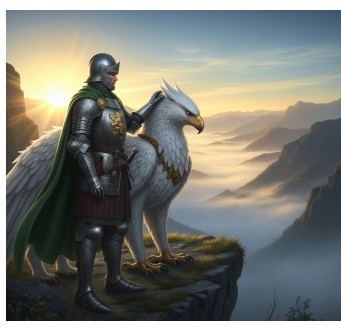 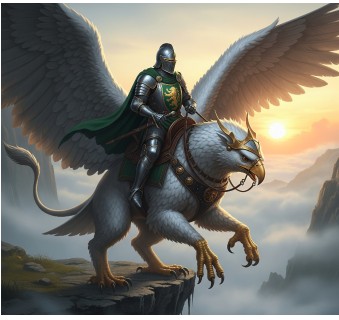 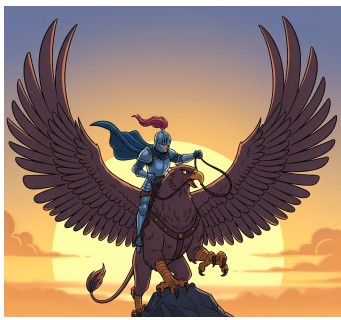

(a) Initial Image for Q1.  (b) Generated with IUT.  (c) Generated without IUT.

Figure 4: Example 1. **Q:** A knight and his griffin companion prepare to set off at dawn. **A:** The knight mounted his griffin, which spread its massive wings, ready to take flight towards the rising sun, its posture full of power.

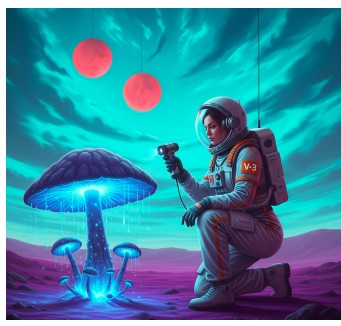 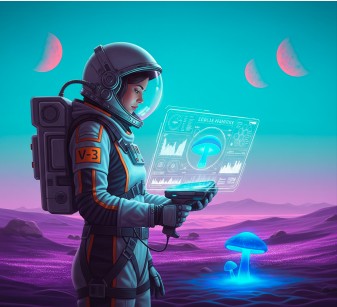 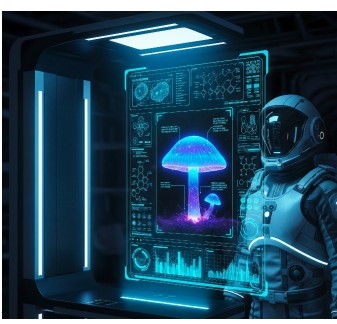

(a) Initial Image for Q2.  (b) Generated with IUT.  (c) Generated without IUT.

Figure 5: Example 2. **Q:** An astronaut discovers glowing plants on an alien planet. **A:** The astronaut stood up, and the scanner in front of her projected a translucent holographic screen displaying complex data about the glowing mushroom.

```
tags, generate multiple sets of prompts;
5.  If there are no pairs of <image> and </image> tags or if
there are similar tags such as (image) or (/image) besides
the tag pairs, also analyze the nearby text to generate
prompts;
6.  Each prompt should correspond to different visual
elements or scenes described in the text;
7.  Note, begin each image's prompt with <image>, without
adding any additional text, explanations, or numbering.
Only output content separated by <image>.  For example:
<image>A majestic mountain range at sunrise.  <image>A
serene lake reflecting the colorful sky; A dense forest
with tall pine trees.
```

## A.6 ILLUSTRATIVE CASES

As shown in the figures below, we select several examples from various categories for demonstration, including the input questions (both images and text) and the outputs of the evaluated models.

### A.6.1 IUT PERFORMANCE EXAMPLES

This section provides qualitative examples comparing the outputs of the interleaved generation task with and without the IUT-Plug.

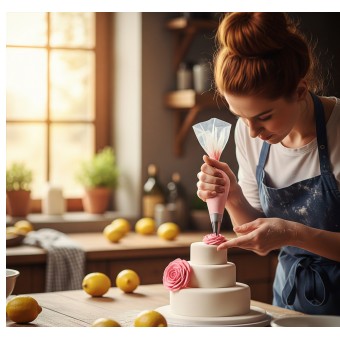 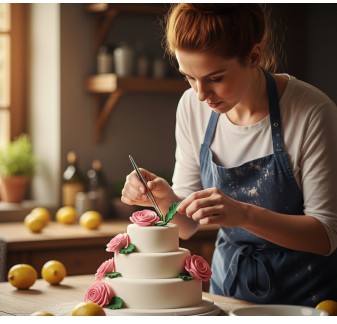 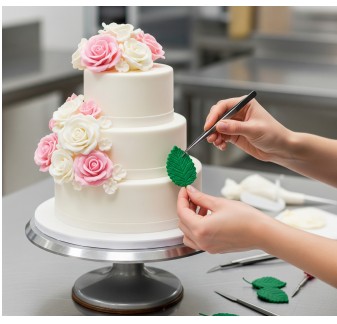

(a) Initial Image for Q3. (b) Generated with IUT. (c) Generated without IUT.

Figure 6: Example 3. **Q:** A female baker is focused on decorating a three-tiered cake. **A:** The three-tiered white cake is now adorned with several roses. She is using tweezers to place a frosting leaf next to a rose, nearing the completion of the cake's decoration.

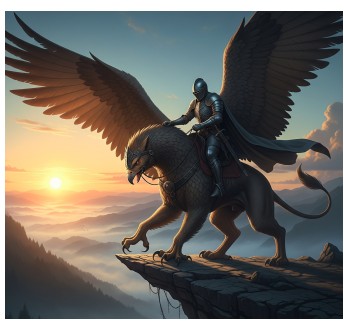 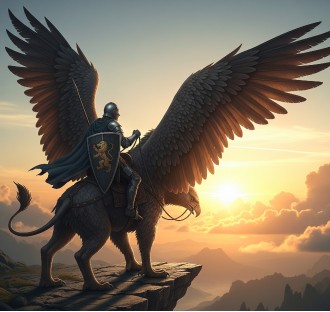 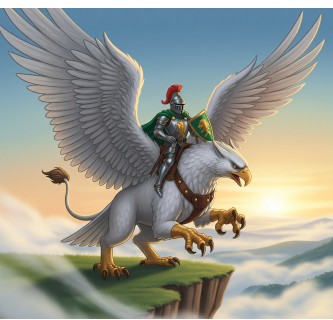

(a) Generated w/o Entity. (b) Generated w/o Relation. (c) Generated w/o Style.

Figure 7: Ablation study for the knight and griffin example. The images show outputs when entity, relation, or style information is omitted from the IUT guidance.

### A.6.2 ABLATION STUDY EXAMPLES

These images support the ablation study, showing the results when key components of the IUT structure (entity, relation, style) are omitted during generation.

### A.6.3 IUT EXTRACTION EXAMPLES

This section provides concrete examples of the structured JSON output generated by the IUT extraction module for given images.

**IUT JSON output for the graduation photo.**

```
{
"global_description":  "The primary subject of a family
posing for a photo at a graduation event features a young
woman in a blue and gold sash, flanked by an older couple
in casual attire under artificial lighting, captured in a
realistic style with warm tones, evoking a sense of pride
and nostalgia.",
"global_features":  {
"style":  "photorealistic",
"lighting":  "soft artificial light",
"...":  "..."
},
"objects":  [
{"name":  "woman wearing glasses", "type":  "person", "...":
"..." },
```

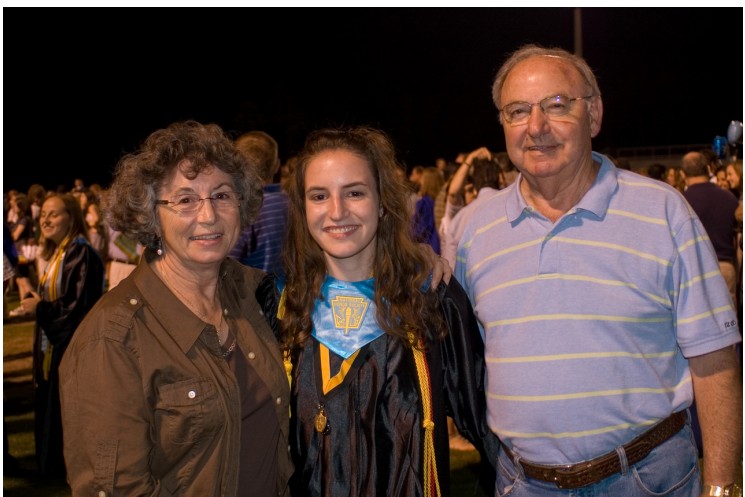

Figure 8: Input image for the first IUT extraction example (Graduation photo).

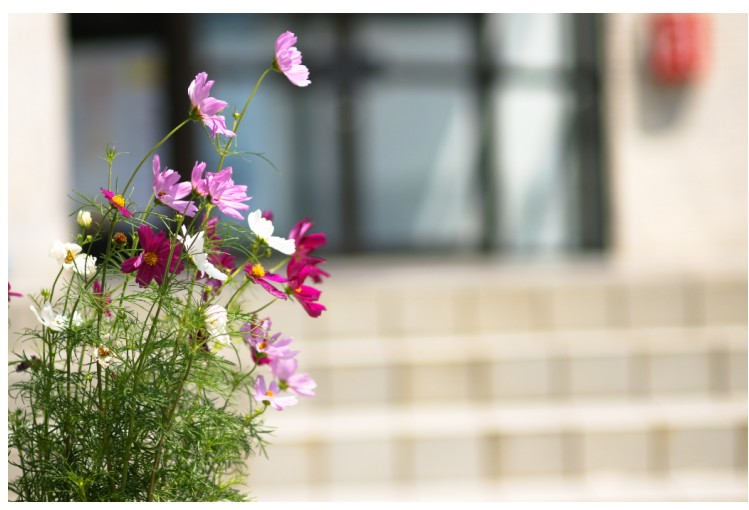

Figure 9: Input image for the second IUT extraction example (Flowers).

```
{"name":  "graduate in cap and gown", "type":  "person",
"...":  "..." }
],
"relationships":  [
"woman standing next to graduate",
"man standing next to graduate",
"..."
]
}
```

**IUT JSON output for the flowers photo.**

```
{
"global_description":  "The vibrant bouquet of cosmos
flowers in shades of pink, purple, and white, set against
a softly blurred urban backdrop, showcases a realistic and
detailed artistic style that evokes a serene and peaceful
atmosphere.",
"global_features": {
"style":  "photorealistic",
```

```
        "lighting":  "soft natural light",
        "...":  "..."
        },
        "objects":  [
        {"name":  "colorful flowers", "type":  "object", "...":
        "..." },
        {"name":  "green stems and leaves", "type":  "object",
        "...":  "..." }
        ],
        "relationships":  [
        "flowers growing in garden",
        "flowers near building"
        ]
        }
```

### A.7 SIX-POINT GRADING SYSTEM CRITERIA

Table 6: Six-point grading system and evaluation prompts

| Evaluation Dimensions | Key Elements of Prompts |
| --- | --- |
| Text Quality | Evaluate the clarity, grammatical accuracy, and relevance of the text. Check for duplications or irrelevant content. |
| Image Relevance | Assess whether visual elements precisely correspond to textual descriptions, rejecting generic/decorative images. |
| Cross-modal Consistency | Verify seamless integration between text and images, with coherent contextual transitions. |
| Task Completion | Measure the completeness of required actions in project-based tasks (e.g., all steps in tutorials). |
| Innovation | Evaluate originality in narrative approaches and visual storytelling techniques. |
| Harmful Content | Deduct 1 point for violent/offensive material (penalty criterion only). |

### A.8 IUT EXTRACTION MODULE PROMPT

The IUT-Plug's core functionality relies on its ability to extract a hierarchical symbolic scene representation from multimodal inputs. As detailed in Section 3, this extraction is achieved by leveraging a powerful pre-trained Vision-Language Model (VLM), specifically Qwen2.5-VL-72B in our setup, as an intelligent scene parser. The VLM is prompted with the input image and a detailed instruction to output a structured JSON object representing the Image Understanding Tree (IUT). This process is crucial for establishing and dynamically updating the persistent symbolic memory across turns.

Figure 10 illustrates the workflow of the IUT extraction module. The VLM receives the current visual input ($I_t$), the user's instruction ($Q_t$), and crucially, the **previous IUT state** ($\mathcal{M}_{t-1}$) serialized as a JSON string. This allows the VLM to perform an informed, incremental update rather than parsing the scene from scratch.

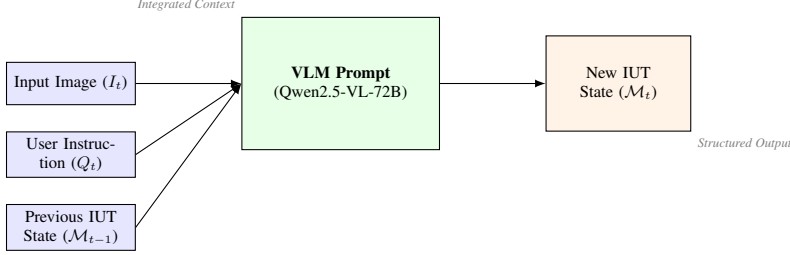

Figure 10: Workflow of the IUT Extraction Module. The VLM is prompted with the current image, user instruction, and the previous IUT state to generate an updated IUT in JSON format.

**Prompt Template for IUT Extraction (Simplified).**

**Role:** You are an expert visual scene parser and knowledge graph constructor. Your task is to analyze the provided image and dialogue history to generate or update a JSON object representing the current Image Understanding Tree (IUT).

**Instructions:**

1. Carefully examine the `<image>` and the `User Instruction`.
2. Consider the `Previous IUT State` to maintain continuity.
3. Output a JSON object that strictly adheres to the schema below.
4. Update existing entities/attributes/relations or add new ones based on the current input.
5. If an entity or attribute is no longer relevant or has changed, reflect that in the updated JSON.

**JSON Schema (Must Follow):**

```
{
"timestamp":  "YYYY-MM-DDTHH:MM:SSZ",
"scene_summary":  "A concise textual description of the overall scene.",
"global_style": {
"artistic_medium":  "e.g., 'photorealistic', 'cartoon', 'watercolor'",
"lighting":  "e.g., 'bright natural light', 'dim indoor lighting'",
"color_palette":  "e.g., 'vibrant', 'monochromatic', 'pastel'"
},
"entities":  [
{
"id":  "unique_entity_id_1",
"name":  "e.g., 'cat', 'red mat'",
"attributes":  {
"color":  "e.g., 'black', 'red'",
"state":  "e.g., 'sleeping', 'running'",
"material":  "e.g., 'wood', 'fabric'"
},
"relationships_to_others":  [
{"target_id":  "unique_entity_id_2", "relation":  "e.g., 'on', 'next to',
'holding'"}
]
}
],
"relationships":  [
"entity_name_A [relation_verb] entity_name_B"
]
}
```

**Current Input Context:**

```
<image>
User Instruction:  [User's current instruction,
e.g., "make the cat stand up"]
Previous IUT State:  [Previous turn's IUT JSON,
e.g., ``{"scene_summary":  "A black cat sleeping on
a red mat.", ...}'']
```
**Output:**

This detailed prompt, combined with the VLM's powerful few-shot learning capabilities, enables robust and consistent IUT construction across complex multi-turn interactions. The structured JSON output ensures that the extracted knowledge is machine-readable and directly usable by downstream generation modules.

